# Evaluating the Effectiveness of Brief Interventions for Smoking Cessation Performed by Family Doctors

**DOI:** 10.3390/medicina60121985

**Published:** 2024-12-02

**Authors:** Sergiu Ioachim Chirila, Catalin Nicolae Grasa, Leonard Gurgas, Constantin-Viorel Cristurean, Loredana Hanzu-Pazara, Tony Hangan

**Affiliations:** Faculty of Medicine, Ovidius University of Constanta, 900470 Constanța, Romania; sergiu.chirila@univ-ovidius.ro (S.I.C.); catalin.grasa@yahoo.com (C.N.G.); cristurean_viorel@yahoo.com (C.-V.C.); loredana.pazara@univ-ovidius.ro (L.H.-P.); tony@medcon.ro (T.H.)

**Keywords:** very brief intervention, brief intervention, smoking cessation, tobacco, quit smoking

## Abstract

*Background and Objectives*: Tobacco smoking is the most important cause of chronic diseases and premature death worldwide. Very brief advice (VBA) and brief advice (BA) represent evidence-based interventions designed to increase quit attempts. These are appropriate for all smokers, regardless of their motivation to quit, and involve several steps regarding the assessment, advice, and action. This review aimed to evaluate the effectiveness of these brief interventions offered by general practitioners (GPs) in smoking cessation. *Materials and Methods*: A systematic search for articles that presented these interventions as an aid to support smoking cessation was conducted. The rate of successful smoking cessation was evaluated following interventions offered by general practitioners, regardless of the patients’ present motivation to quit. *Results*: We have checked if these interventions can be used as an innovative method to help active smokers make an informed decision regarding their behaviour. *Conclusions*: Assisted/supported/guided by a general practitioner, current cigarette smokers can decide to quit smoking and identify the best way of cessation. We processed relevant data where brief interventions were used as the main counselling method to aid smoking cessation, regardless of using nicotine replacement therapy (NRT), heated tobacco products (HTPs), or vaping.

## 1. Introduction

Tobacco smoking has emerged as a major public health concern, contributing to over eight million deaths globally each year [1]. It is one of the most preventable causes of chronic diseases such as chronic obstructive pulmonary disease (COPD), asthma, heart failure, and pulmonary hypertension, all of which are significant contributors to premature mortality worldwide [2]. More than 90% of smoking-related deaths and morbidity are attributed to burn-tobacco products, making them the most hazardous form of tobacco consumption [3]. Preventing smoking initiation and increasing the number of successful smoking cessation attempts among current smokers are critical strategies for enhancing population health [4]. The WHO Framework Convention on Tobacco Control was established in response to the tobacco epidemic, implementing population-level measures, such as legislation and tobacco taxation, and individual-level interventions, such as pharmaceutical and non-pharmaceutical support for smoking cessation [5]. In this context, international projects have been initiated to reduce the burden of non-communicable diseases associated with tobacco use and exposure, particularly among adolescents [6].

Due to the highly addictive nature of smoking, most of the smokers who attempt to quit independently relapse within two months [7]. Studies indicate that only a small fraction of individuals successfully quit smoking without any assistance [8,9]. Primary methods, such as pharmacological aids—including nicotine replacement therapies—and behavioural support, only marginally increase the success rate to less than 25% [10]. Previous research has shown that pharmacological support alone does not improve the success rate by more than 15% [11]. The current situation is problematic and necessitates new strategies for supporting smoking cessation, including but not limited to very brief advice [12].

Studies have demonstrated that, in the long term, smokers that are guided by their general practitioner present higher odds of achieving long-term success in smoking cessation, with lower rates of failure in their quit attempts [11]. Very brief advice (VBA) is an evidence-based intervention designed to increase quit attempts among smokers. This intervention aims to support all smokers, regardless of their current motivation to quit [2,11]. Developed by the National Centre for Smoking Cessation Training (NCSCT) in the United Kingdom [13], VBA involves three steps: Ask, Advise, and Act [11].

The three steps in VBA (3A) are as follows: “Ask” patients to disclose their smoking status, “Advise” them on the best-quitting methods or encourage a quit attempt if they were previously uninterested, whether through a single therapy or a combination of medication and behavioural support, and “Act” by providing support during their smoking cessation attempt using the available therapies and methods [11,14,15].

VBA is grounded in the COM-B model (“capability”, “opportunity”, “motivation”, and “behaviour”) for behaviour change [16]. This model identifies the necessary changes for a behaviour change intervention to be more effective than competing behaviours [11]. As a new method, VBA appears to be highly effective as an initial step in the process of quitting smoking [2]. General practitioners (GPs) ask patients about their current smoking status (Ask), advise them to quit smoking and collaboratively identify the best cessation method (Advise), and finally, act by offering practical support to patients interested in quitting (Act) [11].

Brief interventions on smoking cessation, conducted in primary care, are effective and cost-efficient measures [17]. Compared to other health interventions, smoking cessation is highly cost-effective, as it significantly reduces the burden on healthcare systems [18]. The availability and effectiveness of specialized and intensive smoking cessation services vary by country and policy; in some regions, these services are well-established and play a crucial role in public health, while in others, they are considered too expensive and passive to have a major clinical and public health impact [19,20,21]. In contrast, very brief advice is easy to implement at the individual level, requiring minimal training for general practitioners (GPs) and only a few moments to advise patients (generally less than 30 s) [2,11]. However, VBA is not commonly applied in routine care, and data on its usage and effectiveness are limited. Despite this, VBA has the potential to significantly impact smoking cessation, especially as an initial step in the quitting process [2,21].

This review aims to explore VBA and other brief advice methods usefulness in primary care on smoking cessation, both at individual and population levels.

## 2. Materials and Methods

We searched for studies that evaluate the efficiency of smoking cessation advice provided by general practitioners relative to other smoking cessation interventions, such as no intervention at all, counselling, nicotine replacement therapies, or other medical interventions. In order to obtain a broader perspective, we also used studies that evaluate general practitioner’s opinions and experiences on using brief advice interventions.

We included studies where participants were advised to stop smoking traditional cigarettes, regardless of their current intention to quit. Our evaluation focused on the success rate of smoking cessation when using very brief advice (VBA) as the primary counselling intervention. We concentrated on studies that linked VBA interventions to changes in smoking behaviour. We searched PubMed, Web of Science, Science Direct, and Scopus databases (Table 1). We searched using the following keywords:very brief advice/ brief behavioural counselling;smoking cessation advice/ telephone follow-up;brief interventions/ physician advice;general practice advice/ quit attempts;smoking reduction intervention;smoking cessation/ smoking quitting;harm reduction/ risk assessment;smoking relapse prevention.

We observed that Science Direct yielded the highest number of search results when using simple keywords or syntax, whereas other databases offered more accurate advanced-search capabilities. Given that Science Direct might return an overwhelming number of results, future research should compare article selection across different databases to mitigate potential bias from result selection.

We presented in the following PRISMA 2020 flow diagram for new systematic reviews the steps made in selecting final articles for this review (Figure 1).

We included relevant studies that examine the efficiency of advice interventions for smoking cessation. For self-reports, experts agreed on criteria such as prolonged abstinence from 6 to 12 months, point prevalence abstinence of at least 7 days, and continuous abstinence for a minimum of 6 months [22]. The preferred methods for validation are observational biochemical tests, including carbon monoxide levels from expired air and cotinine levels from saliva. Optimal follow-ups are conducted at 6 to 12 months, with or without intermediate check-ups [22].

### Data Collection and Analysis

We checked the titles and abstracts of articles identified in the search and compared them with our goals; each author participated in the screening process, making sure that each article was evaluated by at least two researchers. For the final selection, we confronted and combined the selection to reach the result. Any disagreements were solved through discussion or consulting another review author.

## 3. Results

After searching PubMed, Web of Science, ScienceDirect, Scopus, etc., for the keywords above, over five hundred eligible articles were found. After applying the inclusion and exclusion criteria, fifteen articles were found eligible for this review.

There is a mix of studies included; some are quantitative studies that directly measure the effects of brief interventions in smoking cessation, while others evaluate the perceptions of the general practitioners’ or patients’ perspectives on using very brief advice.

Of the studies included in the review, ten of them revealed statistically significant results for brief interventions efficiency in smoking cessation. These studies are presented below in Table 2. In the remaining five studies, the data provided showed no statistically significant results or the contrary, meaning less effectiveness for using brief interventions in smoking cessation compared to more traditional smoking cessation aids, such as nicotine replacement therapy (NRT), varenicline, bupropion, cytisine, clonidine, nortriptyline, and escitalopram [22] (Table 3).

## 4. Discussion

The results of the analysis indicate the superior effectiveness of using VBA as a smoking cessation aid in primary care, administered by healthcare workers. A short piece of advice from a doctor appears to be at least as effective as smoking cessation medication [35]. The VBA intervention employs the 3A approach (Ask–Advise–Act) and typically requires less than 30 s, making it much easier to implement than other methods, because they can stop the progression of smoking despite the interaction problems generated in clinical practice [36].

Alateeq, Mohammed Alrashoud, et al. evaluated primary healthcare physicians’ attitudes toward and practice of delivering smoking cessation advice to smokers in a military community in central Saudi Arabia [23]. More experienced physicians and those with a positive attitude were more likely to report engaging in favourable practices compared to their less experienced counterparts or those with a negative attitude. [23]. Studies indicate that doctors’ positive attitudes increase proportionally with their experience, thereby enhancing communication with patients and fostering greater trust [37].

Papadakis, Sophia Anastasaki, et al. examined the VBA intervention for smoking cessation in a cohort of 134 patients, with a mean age of 55.3 years and 50.7% female, of whom 42.5% were current smokers. The study revealed that a significant proportion of smokers believed that willingness alone is both ‘necessary’ and ‘sufficient’ for quitting [11]. Consequently, these patients perceive smoking not as a disease but rather as a vice, making quitting a matter of willpower rather than medical treatment [38]. Since the vast majority of smokers believe that willpower alone is enough for smoking cessation, this paradoxical belief often prevents them from attempting to quit [39]; consequently, only one in four patients (12 out of 50) proved to be willing to schedule an appointment to discuss smoking cessation [11].

Onno C P Bindels, Lynn Nijs, et al. examined the opinions of general practitioners (GPs) regarding VBA and its role in smoking cessation. The study involved 18 GPs and 5285 patients. The findings concluded that GPs hold a highly positive opinion of VBA. All participating GPs expressed intentions to continue using VBA in their daily practice. They found the method to be efficient in terms of time, patient-friendliness, and ease of implementation [2]. This is all the more beneficial for patients who have cancer and for whom quitting smoking is all the more difficult [40].

Spaducci, Gilda Richardson, et al. analysed whether quit attempts increase in frequency after referral to specialist smoking cessation support in the inpatient mental health setting. The study included 5434 unique individuals over 52 months for a four-year period, across four psychiatric hospitals [14]. Smoking culture in a mental health context is a recurring theme in many studies [41]. Smokers with mental health conditions are as motivated, if not more, to quit smoking compared to those without mental health problems [42]. However, these patients are less likely to receive support than the general population [43] due to the prejudice associated with the “therapeutic” function of smoking in this group [44,45].

Cheung YJiang NJiang C et al. investigated the effectiveness of very brief interventions in smoking cessation by analysing the quit rates among Chinese outpatient smokers. A randomized controlled trial was conducted on 13,671 smokers in China. The intention-to-treat analysis revealed that the intervention group exhibited higher self-reported 7-day and 30-day abstinence rates at the 12-month follow-up. The effect size was more significant when only participants who received the intervention from compliant physicians were included. However, the difference between groups in terms of biochemically validated abstinence was minimal [24]. Stead et al. found that the quit rate was significantly higher following brief advice from physicians compared to no advice or usual care [10]. They recommend brief behavioural interventions as a time-efficient method to support smokers in quitting [46].

In their study, Chan, Sophia Siu Chee Cheung et al. evaluated the effectiveness of brief smoking-cessation advice provided by briefly trained youth counsellors during the enrollment phase of an incentive-based smoking cessation campaign. According to the intention-to-treat analysis, the intervention group exhibited a non-significantly higher self-reported quit rate and validated quit rate compared to the control group at the 6-month follow-up. [25]. There are studies focusing on smoking cessation among adolescents from low- and middle-income backgrounds, particularly in the realm of behavioural interventions. These studies emphasize the significant impact of brief messages in promoting smoking cessation [47,48]. These behavioural interventions play a crucial role in facilitating smoking cessation among adolescents [49]. The additional components, beyond the brief advice, have a negligible impact [50].

Li, William H C Wang et al. examined the effectiveness of smoking cessation in a randomized controlled trial using brief advice based on risk communication. A total of 528 cancer patients who smoked were randomly assigned to two groups. At the 6-month follow-up, the intervention group exhibited a higher rate of at least 50% self-reported reduction in smoking compared to the control group [34]. Studies indicate that the longer the duration since smoking cessation, the better the survival outcome [51]. In this context, the motivation to quit smoking is associated with greater efficacy, as perceptions of risk are crucial; most patients do not support fatalistic beliefs [52].

Van Rossem, Carolien Spigt, et al. conducted a study to investigate the effectiveness of intensive counselling by a nurse practitioner (NP) versus brief advice by a general practitioner (GP), each combined with pharmacotherapy, for achieving 6 months of tobacco abstinence (primary outcome) in a cohort of 295 adult daily smokers [26]. Additionally, it has been observed that brief advice from a general practitioner to individuals seeking help to quit smoking is as effective as multiple sessions of behavioural support from a nurse, even when medications are used to aid smoking cessation [26]. However, studies indicate that there is conflicting evidence regarding the effectiveness of physician training in increasing smoking cessation rates [53]. Abstinence rates in the GP group versus the NP group were higher, without being statistically significant, from weeks 9 to 26 and with similar results (not statistically significant) from weeks 9 to 52, respectively. Furthermore, good dosing adherence was significantly higher in the GP group compared to the PN group. This is even more relevant considering the important role that medication for smoking cessation plays in primary care. [54].

Wang, Man Ping Li, et al. compared the efficacy of brief advice for cut-down-to-quit with brief advice for quitting immediately, delivered by trained volunteers, without pharmaceutical treatment (*n* = 1077). Researchers found that CDTQ (cut-down-to-quit) and QI (quit immediately) have similar short-term point prevalence abstinence (PPA) rates. Therefore, long-term follow-up is necessary to gain better understanding on the effect of smoking reduction on abstinence [27,55]. Progressive smoking reduction could be considered an alternative intervention goal among smokers who are unwilling to quit smoking altogether [56]. By intention-to-treat analysis, the two groups showed similar results in terms of self-reported point prevalence abstinence, validated abstinence rate, and quit attempts rate at 6 months. Nevertheless, the CDTQ group exhibited a significantly higher reduction rate compared to the QI group. The overall intervention adherence was suboptimal at 45.4%, with particularly low adherence observed in the CDTQ group (42.3%) [27]. There are studies indicating that at the six-month follow-up, the quit rate in the QI (quit immediately) group was significantly higher than in the CDTQ (cut-down-to-quit) group. This suggests that immediate cessation might be more effective [57].

Wu, Lei He et al. examined the effectiveness of brief physician advice. This was applied together with other four very brief telephone calls to promote smoking cessation (*n* = 181) [28]. Currently, in the United States, there is a highly impactful treatment for smoking cessation that combines internet and telephone support to promote sustained abstinence [58]. At the 12-month follow-up, the number of quitters in the VBA group was higher, although not significantly, compared to the EDA control group. The self-reported 7-day point prevalence quit rate (secondary outcome) in the SRI group was significantly higher than the control group at each follow-up interview [28].

Grech, Joseph Sammut, et al. conducted a study to identify the extent of smoking cessation practices among healthcare professionals [29]. The findings revealed that most professionals reportedly asked patients about smoking (76.3%), provided advice (83.5%), and assessed patients’ readiness to quit (70.5%). However, fewer professionals assisted with cessation efforts (40.9%) or arranged follow-ups (24.2%). Despite their efforts, many professionals found it challenging to encourage clients to quit smoking. Interestingly, former healthcare workers that were smokers were 6.86 times more likely to disagree with this perspective compared to those who had never smoked. Based on the data provided in the last four studies, there were no statistically significant results indicating the superiority of using very brief advice (VBA) in smoking cessation compared to more traditional smoking cessation aids, such as nicotine replacement therapy (NRT), varenicline, bupropion, cytisine, clonidine, nortriptyline, and escitalopram, or extended counselling with regular follow-up intervals [14]. Even though patients are satisfied with the VBA intervention, a significant portion are motivated to attempt smoking cessation after receiving VBA from their family physician [11]; however, whether VBA was used as the sole method for quitting smoking or in addition to other pharmacological or psychological therapies, the effectiveness of VBA did not demonstrate significant superiority in promoting smoking cessation.

Utap, M S Tan, et al. examined the efficacy of a brief intervention for smoking cessation utilizing the ‘5A’ model alongside self-help materials compared to employing self-help materials alone. The tobacco cessation protocol involved conducting a routine assessment of smoking habits, providing advice on the importance of quitting smoking, assessing the smoker’s readiness to quit, setting a quit date, and implementing the necessary steps thereafter [59,60]. At the one-month follow-up, 15 out of 77 participants (19.5%) in the intervention group attempted to quit smoking, while 8 out of 80 (10.0%) in the control group did so. However, the authors concluded that there was no statistically significant difference between the two groups. At the four-month follow-up, 13 out of 58 participants (22.4%) in the intervention group attempted to quit smoking, compared to 9 out of 57 (15.8%) in the control group. However, no statistically significant difference was observed between the two groups. These findings should be interpreted with caution due to the high dropout rate and the potential for bias in the study design. [30].

Chan, Sophia S.C. Wong et al. investigated the effectiveness of brief interventions for smokers participating in the Hong Kong Quit to Win Contest to quit smoking [31]. The abstinence rates in the telephone, SMS, and control groups were 22.2%, 20.6%, and 20.3%, respectively [31]. Instant messaging support via WhatsApp could potentially serve as a promising platform for providing smoking cessation support to smokers [61].

Hjalmarson et al. assessed the effectiveness of two in-hospital smoking cessation interventions: brief advice and extended counselling with follow-up. The study included 770 patients who were daily smokers or had quit smoking within 30 days preceding admission to the hospital. The one-year point prevalence self-reported cessation rate was 22% for brief advice and 28% for extended counselling. This suggests that very brief advice (VBA) was not significantly better than usual care. Furthermore, extended counselling is likely to yield additional benefits, although they are modest and do not justify this intervention for unselected smokers, but rather for those who are motivated [62].

Caponnetto, Pasquale Maglia, et al. investigated whether smoking cessation counselling at participants’ workplaces and very brief advice are effective in facilitating successful smoking cessation. In the study, 656 randomized participants in a workplace received either four sessions of group motivational interviewing or four sessions of very brief advice and were subsequently followed-up for 52 weeks [33]. At the participants’ workplace, individual counselling and personalized text messages have shown positive effects on nicotine dependence and expired carbon monoxide levels at 6 and 12 weeks [63]. The Continuous Quit Rate was higher for the group exposed to smoking cessation counselling compared to the very brief advice group during weeks [33]. When interventions were conduced in groups, the quit rate doubled at the 6-month follow-up; therefore, these types of interventions could be recommended as effective smoking cessation methods [64].

## 5. Limitations

The major limitation of the current study is related to the heterogeneity of the included studies. These employ different protocols and heterogeneous approaches for implementing VBA, resulting in significant variation in how VBA intervention is perceived by participants. There may be insufficiently controlled confounding factors in the studies included in the review, such as concurrent interventions for smoking cessation, which could influence the outcomes. The definition of smoking cessation varies across some studies, complicating the synthesis of the final results.

## 6. Future Development

The development and testing of specific versions of very brief advice (VBA) interventions tailored to the needs and behaviours of users of heated tobacco products or vaping devices might prove useful in reducing the burden of tobacco use, as both conventional cigarettes and electronic cigarettes have the potential to influence the production of reactive oxygen species [65] and influence the health status. At the same time, including tobacco use as a risk estimator in digital health technologies [66] that evaluate the risk of cancer or other diseases should be taken into consideration.

## 7. Conclusions

Very brief advice comprises concise steps that usually take less than 30 s, making it one of the simplest, quickest, and cheapest methods available to assist in smoking cessation.

Most of the evaluated studies provide relevant and significant data to support the use of brief interventions for smoking cessation, reporting good results in smoking cessation (as described by Cheung Y. et al. or Wu Lei et al.) [24,28] or a very positive opinion as described by Onno C. et al. [2].

The positive attitude of physicians using VBA strengthens patients’ confidence in this practice, and this is observed even among outpatients or adolescents if the emphasis is placed on immediate smoking cessation. Additionally, there is potential for using modern communication platforms to support smokers in their efforts to quit. Future research should continue to explore and develop new and innovative methods to enhance smoking cessation success among diverse populations.

Due to smokers perceiving smoking cessation more as a matter of willpower than a medical issue, very brief advice (VBA) interventions are necessary to support smoking cessation.

The positive attitude of physicians using VBA strengthens patients’ confidence in this practice, and this is observed even among outpatients or adolescents, especially if the emphasis is placed on immediate smoking cessation. Additionally, there is potential for using modern communication platforms to support smokers in their efforts to quit. Future research should continue to explore and develop new and innovative methods to enhance smoking cessation success among diverse populations.

Although it does not significantly influence biochemically validated smoking abstinence, VBA is beneficial and well-accepted in medical practice for inducing the idea of smoking cessation. Its integration into a broader management plan, which may include other effective interventions, is crucial to increasing the chances of achieving long-term smoking cessation success.

## Figures and Tables

**Figure 1 medicina-60-01985-f001:**
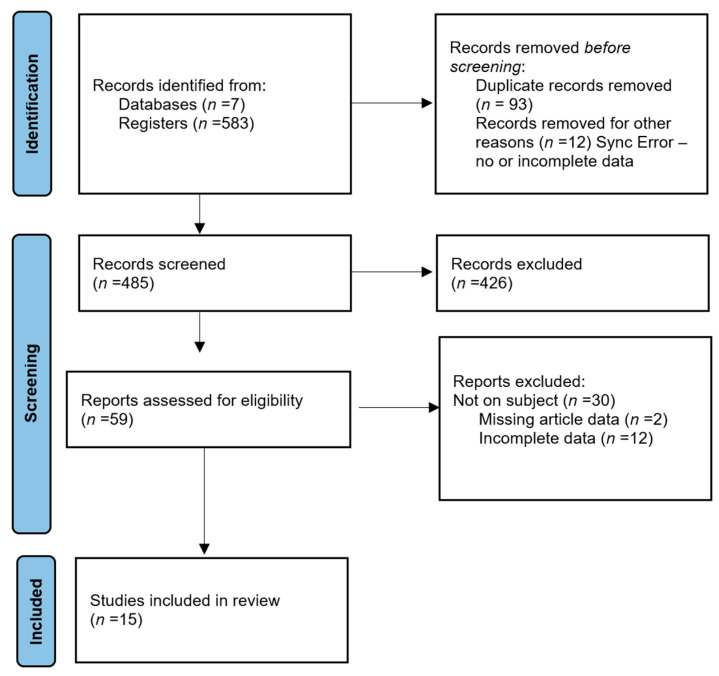
Prisma flowchart of the selection process.

**Table 1 medicina-60-01985-t001:** Advanced-search results.

Keyword/Syntax	PubMed	Science Direct	Web of Science	Scopus
“very brief advice” AND “smoking cessation”	525	4421	21	27
“VBA” AND “smoking cessation”	8	30	7	11
(very brief counselling on smoking cessation) OR (very brief advice on smoke cessation)	1040	4464	76	984
(very brief advice) AND ((smoking quitting) OR (smoking cessation))	528	4968	60	584
((very brief advice) AND (smoking cessation)) AND ((Harm reduction) OR (risk assessment))	37	3171	6	65
((smoking reduction intervention) OR (very brief counselling)) AND (smoking cessation)	4442	24,936	2136	3947
(smoking relapse prevention) AND ((very brief advice) OR (very brief counselling) OR (telephone follow-up))	245	4363	62	75

**Table 2 medicina-60-01985-t002:** Studies that favour brief interventions effectiveness.

Authors	Year	Title	Sample	Objective	Results
Alateeq, Mohammed Alrashoud, Abdulaziz M Khair, Mohammed Salam, Mahmoud et al.	2016	Smoking cessation advice: the self-reported attitudes and practice of primary healthcare physicians in a military community, central Saudi Arabia [23].	73 Primary Healthcare Physicians certified by Saudi Commission for Health Specialties and working within the Riyad region of Saudi Arabia. 71.4% were general practitioners, 17.9% consultants and 10.7% specialists	The study assessed the attitudes and practices of healthcare specialists in providing smoking cessation counselling to smokers within a military base.	All participants agreed that brief smoking cessation advice is part of their job, and more than 90% agreed that the presence of the network of clinics for smoking cessation encouraged them towards this attitude. Totally, 85% of the participants described brief smoking cessation intervention as helpful in quitting. Experience and a positive attitude were identified as factors significantly associated with positive attitude and favourable practice.
Papadakis, Sophia Anastasaki, Marilena Papadakaki et al.	2020	‘Very brief advice’ (VBA) on smoking in family practice: A qualitative evaluation of the tobacco user’s perspective [11].	A total of 134 patients were evaluated, with a mean age of 55.3 years; 50.7% were female, and 42.5% were active smokers. Among the 57 patients identified as current tobacco users, 50 agreed to participate in the study, resulting in a response rate of 87.7%.	The purpose of this study was to evaluate VBA intervention for smoking cessation.	The use of VBA was favourably received by most of the smokers. The motivational impact of receiving advice from a specialist in a supportive and empathic manner was highly praised. Also, the study revealed that a high proportion of patients (60–78%) consider that willpower is a must and is also sufficient to stop smoking. Following the initial VBA intervention, 24% of the patients took steps towards smoking cessation by making appointments at a general practitioner.
Onno C P Bindels, Lynn Nijs, Ancka van Engelen et al.	2020	The experience of general practitioners with Very Brief Advice in the treatment of tobacco addiction [2].	18 GPs took part in the study/5285 patients	The primary objective of the study was to assess the perspectives of the GPs on the use of VBA and its effectiveness in supporting smoking cessation.	The study concluded that GPs had a positive attitude towards applying VBA and had a good experience. They perceived the method as time-efficient, patient-centered, and straightforward to implement, making it a valuable tool for promoting smoking cessation. At the same time, interviewed GPs evaluated that their patients had a positive attitude when VBA was used, compared to other methods.
Spaducci, Gilda Richardson, Sol McNeill, Ann Pritchard et al.	2020	An observational study of system-level changes to improve the recording of very brief advice for smoking cessation in an inpatient mental health setting [14].	5434 unique individuals during the study period; four psychiatric hospitals; over 52 months (May 2012–September 2016)	The study evaluated the influence of different system-level changes with the purpose of improving the process of using and reporting VBA and referral to specialist smoking cessation support in patients from mental health setting.	From a total of 8976 admissions, approximately 57% of the patients were recorded as smokers. Of these smokers, 78.2% received advice, and 79.2% received an offer of a referral to specialised smoking cessation service. From the patients that were offered a referral, 26.6% agreed to seek specialised help.
Cheung YJiang NJiang C et al.	2021	Physicians’ very brief (30-s) intervention for smoking cessation on 13,671 smokers in China: a pragmatic randomized controlled trial [24].	The study evaluated adults who were daily cigarette smokers, (13,671 individuals), with 99% of them being male.	The researchers investigated the impact of very brief advice (VBA) interventions on smoking cessation rates among Chinese outpatient smokers.	In an intention-to-treat analysis, the intervention group demonstrated higher self-reported 7-day abstinence rates compared to the control group (9.1% vs. 7.8%, odds ratio [OR] = 1.14, 95% confidence interval [CI] = 1.03–1.26, *p* = 0.008) and 30-day abstinence rates (8.0% vs. 6.9%, OR = 1.14, 95% CI = 1.03–1.27, *p* = 0.01) at the 12-month follow-up.
Chan, Sophia Siu Chee Cheung, Yee Tak Derek Wong et al.	2018	A Brief Smoking Cessation Advice by Youth Counselors for the Smokers in the Hong Kong Quit to Win Contest 2010: a Cluster Randomized Controlled Trial [25].	A total of 831 adult smokers participated in the study. 441 were part of the intervention group (5 min smoking cessation advice session from young counsellors) and 390 participants were not exposed to this intervention.	The study assessed the effectiveness of brief smoking cessation advice during the enrolment phase of an incentive-based smoking cessation campaign.	Participants exposed to brief advice self-reported, in a higher proportion, smoking abstinence at 6 month follow-up compared, to participants from the control group (18.4% vs 13.8%).
van Rossem, Carolien Spigt, Mark Viechtbauer	2017	Effectiveness of intensive practice nurse counselling versus brief general practitioner advice, both combined with varenicline, for smoking cessation: a randomized pragmatic trial in primary care [26].	The study was conducted on 295 daily smokers. Participants were randomized to receive intensive individual counselling by a practice nurse (PN) or brief intervention by a GP.	The study evaluated the effectiveness of intensive smoking cessation interventions conducted by practice nurse versus brief advice offered by GP in smoking abstinence over a six-month period, on top of pharmacological intervention.	Higher abstinence rates were observed in GP groups at each evaluation (26 weeks and 52 weeks). At the same time, patients on the GP group had a higher adherence rate to the treatment (62% for GP group versus 45.5% for PN group). Thus, a conclusion of the study was that a brief intervention conducted by a GP is at least as effective as intensive support from a specialised practice nurse.
Wang, Man Ping Li, William H. Cheung et al.	2017	Brief Advice on Smoking Reduction versus Abrupt Quitting for Smoking Cessation in Chinese Smokers: A Cluster Randomized Controlled Trial [27].	A total of 1077 smokers were enrold in a contest, with the intention to quit or reduce smoking. These were randomized into two groups, one designed as brief advice about cut-down-to-quit method and the second group randomized to quit immediately.	This study aimed to compare the efficacy of brief advice on cut-down-to-quit (CDTQ) versus brief advice on quitting immediately (QI), both delivered by trained volunteers, in outreach-recruited smokers in Hong Kong who intended to quit smoking. The intervention did not involve the use of pharmacological therapy.	The two groups had similar results at 6-month time point. The cut-down-to-quit group had a higher reduction rate in tobacco consumption compared to quit-immediately group (20.9% vs. 14.5%). Overall intervention adherence was suboptimal at 45.4%, with particularly low adherence in the CDTQ group (42.3%).
Wu, Lei He, Yao Jiang et al.	2017	Very brief physician advice and supplemental proactive telephone calls to promote smoking reduction and cessation in Chinese male smokers with no intention to quit: a randomized trial [28].	Participants were randomized into two groups: 181 participants that received brief physician advice to reduce smoking by at least half within one month, with telephone follow-up and 188 participants that received brief advice on physical activity and healthy diet and telephone follow-up.	The authors investigated the effectiveness of brief advice provided by a physician combined with four brief telephone calls in promoting smoking cessation. This approach was compared to equivalent advice on diet and exercise.	The intention-to-treat analysis revealed that the self-reported 6-month prolonged abstinence rate at the 12-month follow-up was higher in the Smoking-Reduction Intervention group (15.7%) compared to the Exercise and Diet Advice control group (7.8%). The self-reported 7-day point prevalence quit rate, was significantly higher in the SRI group compared to the control group at each follow-up interview. At the 12-month follow-up, the quit rate was 13.3% in the SRI group versus 6.9% in the control group, with an OR of 2.09 (95% CI: 1.01–4.34, *p* = 0.049).
Grech, Joseph Sammut, Roberta Buontempo et al.	2020	Brief tobacco cessation interventions: Practices, opinions, and attitudes of healthcare professionals [29].	Healthcare professionals (*n* = 133) that were trained in using brief interventions for smoking cessation.	This study aimed to identify the extent of smoking cessation practices of healthcare professionals interested in tobacco cessation, and their opinions and attitudes.	Most of the healthcare professionals report applying part of the brief interventions in higher percentages (ask, advice, assess) compared to assist and arrange. The participants concluded that influencing the patients to quit smoking is difficult, former smokers were more positive on towards the outcome.

**Table 3 medicina-60-01985-t003:** Studies that do not favour brief interventions.

Authors	Year	Title	Sample	Objective	Results
Utap, M S Tan, Cpl Su, A T	2019	Effectiveness of a brief intervention for smoking cessation using the 5A model with self-help materials and using self-help materials alone: A randomized controlled trial [30].	The study was conducted on 208 participants, who were randomized into two groups (intervention and control).	The study compared brief interventions based on 5A model with self-help materials compared to self-help materials alone.	At 1-month follow-up, 19.5% of the participants in the intervention group had attempted to quit smoking compared to 10.0% in the control group. Even though at 1 month the intention to quit was higher in the intervention group, at 6 months, there was no statistically significant difference (*p* = 0.37).
Chan, Sophia S.C. Wong, David C.N. Cheung et al.	2014	A block randomized controlled trial of a brief smoking cessation counselling and advice through short message service on participants who joined the Quit to Win Contest in Hong Kong [31]	The 1003 participants were allocated to three groups (TEL group, SMS group, or Control group), with 338, 335, and 330 participants, respectively.	The present trial examined the effectiveness of brief intervention delivered through different methods. The Tel group received a 5 min telephone, the SMS group received eight messages, and the control group was not exposed to any interventions.	At 12-month follow-up, no statistically significant difference was observed between the three groups (Tel—22.2%, SMS–20.6%, Control—20.3%). Nonetheless, when modelling for 2-, 6 -, and 12-months follow-up, a significant difference between TEL group and control group showed.
Hjalmarson, Agneta Boëthius, Göran	2007	The effectiveness of brief advice and extended smoking cessation counselling programmes when implemented routinely in hospitals [32].	The subjects (*n* = 770) were in-patients who were daily smokers or had stopped smoking 30 days preceding admission to 15 wards in five Swedish hospitals.	To evaluate the effectiveness of two in-hospital smoking cessation interventions—brief advice and extended counselling with follow-up	The prevalence at one year was 28% for extended counselling and 22% for brief advice. Difference was not statistically significant. Brief advice was declined by 17% of patients, while 34% refused extended counselling. Of those who received the interventions, only half of the brief advice group and one-third of the counselling group completed the full programme, with cessation rates of 25% and 38%, respectively, among these participants.
Caponnetto, Pasquale Maglia, Marilena Floresta et al.	2020	A randomized controlled trial to compare group motivational interviewing to very brief advice for the effectiveness of a workplace smoking cessation counselling intervention [33].	A total of 656 participants were randomized to receive four sessions of group motivational interviewing or four sessions of VBA and were followed-up for one year.	This study aims to assess smoking cessation counselling at the participants’ workplace and very brief advice on effectiveness in successfully quitting	The Continuous Quit Rate was lower for the very brief advice group compared to smoking cessation counselling group during weeks 9 to 12 (3.6% vs 17.5%) weeks 9 to 24 (3.4% vs 13.4%) and weeks 9 to 52 (3.1% vs 10.3%).
Li, William H C Wang, M P Ho, K Y Lam et al.	2018	Helping cancer patients quit smoking using brief advice based on risk communication: A randomized controlled trial [34].	A total of 528 smoking-cancer patients were randomly allocated either into an intervention group (*n* = 268) to receive brief advice based on risk communication by a nurse counsellor or a control group (*n* = 260) to receive standard care	This randomized controlled trial aimed to examine the effectiveness of a smoking cessation intervention using a risk communication approach.	The study demonstrated that smoking cessation intervention offered better results compared to VBA, with significantly higher percentages of participants that quit smoking.

## Data Availability

Not applicable.

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
