# Peer review of "Evaluating the Effectiveness of Brief Interventions for Smoking Cessation Performed by Family Doctors"

_medicina, 2024, doi:10.3390/medicina60121985_

Round 1
Reviewer 1 Report
Comments and Suggestions for Authors
The manuscript by Chirila et al titled “Evaluating the effectiveness of brief interventions for smoking cessation performed by family doctors” is a systematic review of the literature on very brief advice on smoking cessation delivered by family physicians. The authors have searched PubMed, Web of Science, Science Direct and Scopus databases using search terms that are clearly presented in Table 1. Over 500 eligible articles were identified and ultimately 15 were included in the review. 10 of the articles revealed statistically significant results for the efficacy of brief interventions in smoking cessation; five studies showed no statistically significant result.
Each of the 15 studies is described in considerable detail in tables 2 and 3 which describe of those studies which were statistically significant (Table 2) and not significant (Table 3)
The studies vary considerably in design and purpose but although quite heterogeneous they contain significant nuggets of information that make for a useful summary of strategies for effectively implementing very brief advice in a primary care setting.
The paper is well organized and well written with a few exceptions. The paper is excessively long as there is considerable overlap in the material presented in the tables and in the text. The authors should reduce the redundancies in the manuscript.
The conclusions should highlight those studies that identify the best evidence for the effectiveness of brief interventions performed by family doctors
The paragraph describing the Grech et al study needs to be described with greater clarity. It is not clear what is meant by “former smokers were more likely to disagree compared to those who never smoked (line 278). The study is also referenced inaccurately in table 2 as reference 30 when it is listed as reference 60 in the text
Minor suggestions:
The authors use several acronyms to describe very brief advice including the VBA, BI and BA. The latter two acronyms are not spelled out in full and the reader is left guessing what they stand for. Consistent use of VBA would be preferable.
Spelling mistakes are noted in line 130 and 138
Author Response
Thank you very much for taking the time to review this manuscript. Please find the detailed responses below and the corresponding revisions/corrections highlighted/in track changes in the re-submitted files.
Comment 1: The manuscript by Chirila et al titled “Evaluating the effectiveness of brief interventions for smoking cessation performed by family doctors” is a systematic review of the literature on very brief advice on smoking cessation delivered by family physicians. The authors have searched PubMed, Web of Science, Science Direct and Scopus databases using search terms that are presented in Table 1. Over 500 eligible articles were identified and ultimately 15 were included in the review. 10 of the articles revealed statistically significant results for the efficacy of brief interventions in smoking cessation; five studies showed no statistically significant result.
Each of the 15 studies is described in considerable detail in tables 2 and 3 which describe of those studies which were statistically significant (Table 2) and not significant (Table 3)
The studies vary considerably in design and purpose but although quite heterogeneous they contain significant nuggets of information that make for a useful summary of strategies for effectively implementing very brief advice in a primary care setting.
Response 1: Thank you for your feedback. We greatly appreciate the time and effort you dedicated to reviewing our work. Following your suggestions, we believe the clarity of the manuscript has increased significantly.
Comment 2: The paper is well organized and well written with a few exceptions. The paper is excessively long as there is considerable overlap in the material presented in the tables and in the text. The authors should reduce the redundancies in the manuscript.
Response 2: Thank you for this suggestion. We reevaluated the text and removed parts of the redundancies to obtain a shorter and easier-to-follow manuscript
Comment 3: The conclusions should highlight those studies that identify the best evidence for the effectiveness of brief interventions performed by family doctors
Response 3: Thank you for this advice, we added one more paragraph highlighting relevant studies with positive results.
Comment 4: The paragraph describing the Grech et al study needs to be described with greater clarity. It is not clear what is meant by “former smokers were more likely to disagree compared to those who never smoked (line 278). The study is also referenced inaccurately in Table 2 as reference 30 when it is listed as reference 60 in the text
Response 4: Thank you for this valuable observation. We rephrased the text for clarity. Also, following this observation, we reevaluated all the references to ensure there were no other inaccuracies.
Minor suggestions
Comment 5: The authors use several acronyms to describe very brief advice including the VBA, BI and BA. The latter two acronyms are not spelled out in full and the reader is left guessing what they stand for. Consistent use of VBA would be preferable.
Response 5: Thank you for your feedback. We have revised sections of the text to define the acronyms more explicitly and clearly. The entire text has been reevaluated to ensure proper use of acronyms and clarity for the reader.
Comment 6: Spelling mistakes are noted in line 130 and 138
Response 6: Thank you for your feedback. We corrected the spelling mistakes and, considering the revision, we rechecked the whole manuscript.
Reviewer 2 Report
Comments and Suggestions for Authors
This manuscript represents a review of the effectiveness of brief interventions (BI) for smoking cessation performed by family doctors, and especially the Very Brief Advice (VBA) approach. The manuscript is well written. The search parameters were appropriate. The treatment of the topic was balanced and included both reports that supported and reports that did not support the BI/VBA approach. Limitations were stated. I have only a few comments to be addressed for minor revision:
1. Line 20: Should be: "VBA or brief interventions (BI).
2. Line 21 should be: nicotine replacement therapy (NRT), possibly heated tobacco products (???) (HTPs).
3. Line 59: Should BA be VBA?
4. Line 84: The redundant phrase, "...no intervention of..." should be deleted.
5. Line 231: "practice nurse (PN)" should probably be "nurse practitioner (NP).
6. Lines 242-244: A statement in Romanian should be rephrased in English.
Author Response
Comment: This manuscript represents a review of the effectiveness of brief interventions (BI) for smoking cessation performed by family doctors, and especially the Very Brief Advice (VBA) approach. The manuscript is well written. The search parameters were appropriate. The treatment of the topic was balanced and included both reports that supported and reports that did not support the BI/VBA approach. Limitations were stated. I have only a few comments to be addressed for minor revision:
Response: Thank you very much for taking the time to review this manuscript. Please find the detailed responses below and the corresponding revisions/corrections highlighted/in track changes in the re-submitted files.
Comment 1: Line 20: Should be: "VBA or brief interventions (BI).
Response 1: Thank you for your suggestion. We rephrased the text for a better understanding of the text.
Comment 2: Line 21 should be: nicotine replacement therapy (NRT), possibly heated tobacco products (???) (HTPs).
Response 2: Thank you for the suggestion. We rephrased the text and believe that now it offers a more precise and easier-to-understand text.
Comment 3. Line 59: Should BA be VBA?
Response 3: Thank you for pointing out this inaccuracy. Indeed, it is VBA and we corrected the text accordingly.
Comment 4. Line 84: The redundant phrase, "...no intervention of..." should be deleted.
Response 4: Thank you for this observation. We deleted the redundant words.
Comment 5. Line 231: "practice nurse (PN)" should probably be "nurse practitioner (NP).
Response 5: Thank you for the suggestion. Indeed, nurse practitioner was the intended terminology to use, and we changed in text accordingly.
Comment 6. Lines 242-244: A statement in Romanian should be rephrased in English.
Response 6: Thank you for the observation. We have rephrased it in English.